# C1 neurons are part of the circuitry that recruits active expiration in response to the activation of peripheral chemoreceptors

Milene R Malheiros-Lima[1], Josiane N Silva[2], Felipe C Souza[2], Ana C Takakura[2], Thiago S Moreira[1]*

[1]Department of Physiology and Biophysics, Institute of Biomedical Science, University of São Paulo, São Paulo, Brazil; [2]Department of Pharmacology, Institute of Biomedical Science, University of São Paulo, São Paulo, Brazil

**Abstract** Breathing results from the interaction of two distinct oscillators: the pre-Bötzinger Complex (preBötC), which drives inspiration; and the lateral parafacial region (pFRG), which drives active expiration. The pFRG is silent at rest and becomes rhythmically active during the stimulation of peripheral chemoreceptors, which also activates adrenergic C1 cells. We postulated that the C1 cells and the pFRG may constitute functionally distinct but interacting populations for controlling expiratory activity during hypoxia. We found in rats that: a) C1 neurons are activated by hypoxia and project to the pFRG region; b) active expiration elicited by hypoxia was blunted after blockade of ionotropic glutamatergic receptors at the level of the pFRG; and c) selective depletion of C1 neurons eliminated the active expiration elicited by hypoxia. These results suggest that C1 cells may regulate the respiratory cycle, including active expiration, under hypoxic conditions.

**\*For correspondence:** tmoreira@icb.usp.br

## Introduction

Physiological systems interact in order to promote survival in the face of environmental, metabolic and behavior challenges. Understanding how the neuronal network interacts to control these physiological systems is essential for comprehending our ability to promote the appropriate responses and avoid failures in the face of these challenges. One of the physiological systems that is essential to maintain homeostasis is the respiratory system. Therefore, the process of breathing is a complex and dynamic behavior divided into three phases (inspiration, post-inspiration and active expiration), whereby respiratory muscles produce a pressure difference allowing airflow into and out of the lungs (*Richter and Smith, 2014*). These three phases are the results of the recruitment of different muscles (in general: diaphragm, upper airways and abdominal muscles). Their rhythms are controlled by two distinct oscillators: the pre-Bötzinger Complex (preBötC), which drives inspiration; and the lateral parafacial region (pFRG), which drives active expiration (*Ramirez and Baertsch, 2018*). There is a hypothesized third oscillator located at the post-inspiratory complex (PiCO) which drives the post-inspiratory phase of the breathing cycle (*Del Negro et al., 2018*).

Respiration influences the cardiovascular system through two different mechanisms: Traube-Hering waves, which are related to oscillations in arterial pressure, and the respiratory sinus arrhythmia, which is related to oscillation in the heart rate (*Traube, 1865*; *Hering, 1869*; *Anrep et al., 1936*; *Machado et al., 2017*). The respiratory oscillators are located in the ventrolateral medulla, intermingled with neurons that are involved with cardiovascular control (*Guyenet, 2006*; *Richter and Smith, 2014*; *Guyenet, 2014*; *Machado et al., 2017*; *Del Negro et al., 2018*), suggesting that they can interact to coordinate respiratory and cardiovascular adjustments.

The rostral ventrolateral medulla (RVLM) contains many types of neurons that regulate the sympathetic and parasympathetic outflows as well as breathing output (*Guyenet, 2006*; *Guyenet, 2014*; *Del Negro et al., 2018*). Respiratory physiologists have named at least four functional segments within the RVLM, not counting the parafacial respiratory group/retrotrapezoid nucleus, which can be viewed as the rostral-most extension of the so-called ventral respiratory column (VRC) (*Guyenet and Bayliss, 2015*; *Del Negro et al., 2018*), whereas cardiovascular physiologists have named two groups of neurons in the RVLM that are involved in sympathetic control: catecholaminergic C1 neurons and non-C1 neurons. In the so-called 'cardiovascular/sympathetic' area of the RVLM, catecholaminergic C1 neurons were first identified more than 40 years ago (*McAllen and Dampney, 1990*; *Guyenet, 2006*; *Guyenet et al., 2013*). C1 neurons have distinct projections to the whole brain, demonstrating that these neurons are involved in more than just cardiovascular regulation, as had been previously described (*Ross et al., 1981*; *Li and Guyenet, 1996*; *Schreihofer and Guyenet, 1997*; *Ritter et al., 1998*; *Marina et al., 2011*; *Abbott et al., 2013b*). Selective stimulation of C1 cells causes a rise in arterial pressure and breathing activity (*Abbott et al., 2013b*; *Burke et al., 2014*; *Malheiros-Lima et al., 2018a*). These effects mimic the cardiovascular and respiratory responses elicited by hypoxia (*Burke et al., 2014*). There is evidence that C1 cells are highly collateralized, overlap with respiratory neurons in the VRC and contribute to hypoxic responses via connections with pontomedullary structures (*Guyenet, 2006*; *Guyenet, 2014*).

Hypoxia is considered a very potent stimulus that produces different physiological perturbations, including breathing activation (*Guyenet, 2000*; *Prabhakar and Semenza, 2015*). During a hypoxic challenge, expiration (which is passive at rest) turns into an active process, with dilatation of the upper airways and the recruitment of abdominal expiratory muscles during the second phase of the expiratory process (*Iscoe, 1998*). A group of expiratory neurons located in the parafacial respiratory group (pFRG) has been considered to be a conditional expiratory oscillator (*Janczewski and Feldman, 2006*; *Pagliardini et al., 2011*). The pFRG is located ventrolateral to the facial motor nucleus and becomes rhythmically active when it is necessary to increase lung ventilation, for example during physical exercise or during the stimulation of central and/or peripheral chemoreceptors (*Pagliardini et al., 2011*; *Huckstepp et al., 2015*; *Korsak et al., 2018*).

Therefore, we postulate that adrenergic C1 neurons are in synaptic contact with the pFRG region to recruit expiratory muscles via glutamatergic signaling in order to regulate breathing during hypoxia properly. This notion stems from the following three observations: a) we anatomically verified that C1 neurons are activated by hypoxia and project to the pFRG region; b) active expiration elicited by hypoxia was blunted after blockade of ionotropic glutamatergic receptors at the level of the pFRG; and c) selective depletion of C1 neurons eliminated the active expiration elicited by hypoxia.

## Results

### Active expiration elicited by activation of the parafacial respiratory group is mediated by ionotropic glutamatergic receptors

Under resting conditions, expiration occurs passively as a consequence of the passive deflation of the lung and chest wall to a resting state from a stretched position at the end of inspiration (*Figure 1D–E and H*). Physiologically, active expiration (high ventilatory demand) corresponds to an increase in the airflow in the late expiratory phase, immediately before inspiration, that is due to the recruitment of expiratory muscles. According to others, active expiration is present when the abdominal (Abd$_{EMG}$) amplitude during the final 20% of the expiratory period is >50% larger than that in the first half (*Figure 1D–E*) (*Pagliardini et al., 2011*; *Zoccal et al., 2018*; *Silva et al., 2019*). In adult rats, the pFRG region (a conditional expiratory oscillator) is inactive at rest and can became rhythmically active as a result of changes in the balance of inhibition and excitation, which recruit abdominal muscles by local disinhibition or activation (*Janczewski and Feldman, 2006*; *Pagliardini et al., 2011*; *Huckstepp et al., 2015*; *Huckstepp et al., 2016*; *Huckstepp et al., 2018*; *Zoccal et al., 2018*; *Silva et al., 2019*). Considering the ideas that pFRG neurons are involved in active expiration and that C1 neurons use glutamate as the main neurotransmitter, we began by confirming that the activation of ionotropic glutamatergic receptors in pFRG neurons is able to generate active expirations (*Huckstepp et al., 2018*).

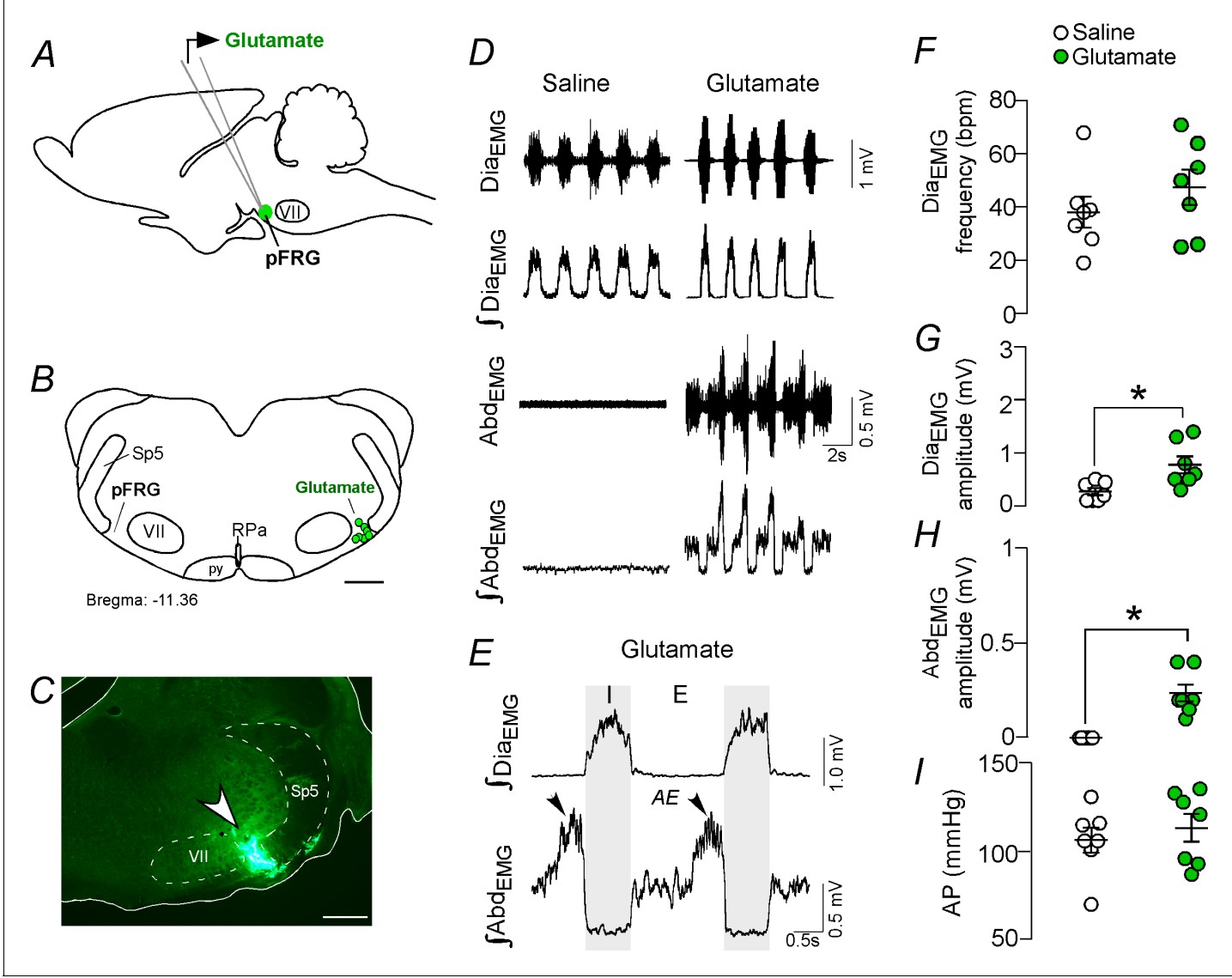

**Figure 1.** Activation of glutamatergic receptors in the pFRG region evoked active expiration. (A) Experimental design. (B) Injection sites of glutamate (10 mM, 50 nL) into the pFRG region. (C) Photomicrography showing a typical injection site of glutamate into the pFRG region. (D) Traces showing breathing parameters (diaphragm electromyography [$Dia_{EMG}$] and abdominal electromyography [$Abd_{EMG}$]) from a representative experiment after the injection of saline or glutamate into the pFRG. (E) Expanded traces after glutamate injection into the pFRG. Black arrows show active expiration (AE) in the expanded traces. (F–I) Individual data for each parameter — (F) $Dia_{EMG}$ frequency (bpm), (G) $Dia_{EMG}$ amplitude (mV), (H) $Abd_{EMG}$ amplitude (mV) and (I) AP (mmHg) — after unilateral injections of saline (white circles; N = 7) or glutamate (green circles; N = 7) into pFRG. Lines show mean and SEM. *Different from saline, Paired Student's t-test, p<0.05. Abbreviations: $Abd_{EMG}$, abdominal electromyogram; AE, active expiration; AP, arterial pressure; bpm, breaths per minute; $Dia_{EMG}$, diaphragm electromyogram; pFRG, parafacial respiratory group; py, pyrimidal tract; RPa, raphe pallidus; Sp5, spinal trigeminal tract; VII, facial motor nucleus. Scale bar in B = 1 mm and C = 0.5 mm.

We performed unilateral injections of glutamate (10 mM, 50 nL; N = 7) (*Figure 1A–C*), followed by ipsilateral injections of the broad spectrum ionotropic glutamatergic antagonist kynurenic acid (Kyn, 100 mM, 50 nL; N = 7) into the pFRG region. According to our histological analysis, the injections were located in the ventral aspect of the lateral edge of the facial nucleus, juxtaposed to the spinal trigeminal tract (*Figure 1B–C*). Unilateral injection of glutamate into the pFRG triggered active expiration, as noted by bursts of $Abd_{EMG}$ during late expiration (*Figure 1D–E*). For example, glutamate injected into the pFRG elicited an increase in $Dia_{EMG}$ amplitude (0.771 ± 0.16 mV vs. baseline of 0.264 ± 0.068 mV; p<0.05) and $Abd_{EMG}$ (0.236 ± 0.045 mV vs. baseline of 0.000 ± 0.000 mV; p<0.05) (*Figure 1D–E and H*). The injection of glutamate into the pFRG did not evoke an

increase in the $Dia_{EMG}$ frequency ($47 \pm 7$ bpm vs. baseline of $38 \pm 6$ bpm, p>0.05) or blood pressure ($127 \pm 8$ mmHg vs. baseline of $120 \pm 7$ mmHg; p>0.05) (*Figure 1D–I*). Previous ipsilateral injection of Kyn into the pFRG completely abolished the active expiration generated by glutamate injections ($0.000 \pm 0.000$ mV vs. $0.236 \pm 0.045$ mV with glutamate but no Kyn; p<0.05). Blockade of the iono-tropic glutamatergic receptors at the level of the pFRG did not affect the resting $Dia_{EMG}$ frequency ($38 \pm 5$ bpm vs. $47 \pm 7$ bpm with active glutamatergic receptors; p>0.05), $Dia_{EMG}$ amplitude ($0.314 \pm 0.114$ mV vs. $0.771 \pm 0.16$ mV with active glutamatergic receptors; p>0.05) or blood pres-sure ($124 \pm 7$ mmHg vs. $127 \pm 8$ mmHg with active glutamatergic receptors; p>0.05).

## Catecholaminergic and glutamatergic terminals at the level of the parafacial respiratory group

Based on the fact that activation of ionotropic glutamatergic receptors elicits active expiration (*Figure 1*; *Huckstepp et al., 2018*) and because a glutamatergic and/or catecholaminergic mecha-nism exists for generating active expiration during hypoxia (*Malheiros-Lima et al., 2017*; *Malheiros-Lima et al., 2018a*), our second series of experiments investigated the presence of catecholaminer-gic innervation in the region of the pFRG. Coronal brainstem sections were immunolabeled for tyro-sine hydroxylase (TH) and vesicular glutamate transporter (VGlut2), two markers that identify catecholaminergic and glutamatergic fibers and terminals in the pFRG region, respectively (*Figure 2A–C*). Catecholaminergic and/or glutamatergic terminals and fibers were observed lateral to the caudal edge of the facial nucleus, a region described as a conditional expira-tory oscillator (*Figure 2A–E*). Within the pFRG, we observed several neurons (NeuN$^+$) in close con-tact with TH$^+$ and/or VGlut2$^+$ terminals located lateral to the caudal tip of the facial nucleus (*Figure 2B*, *Figure 2—figure supplement 1*).

## Parafacial respiratory group targeted by ChR2-expressing catecholaminergic C1 neurons

To specify better the projection from catecholaminergic C1 neurons to the pFRG region, the next series of experiments was designed to identify the presence of axonal varicosities in the pFRG region. Injections of the lentivirus PRSx8-ChR2-eYFP into the left C1 region produced intense fluo-rescence protein expression only in neurons (*Figure 3A*, 3C and 3C'). These neurons (TH$^+$/eYFP$^+$) were always located in close proximity to the original injection site and were concentrated in the region of the ventrolateral reticular formation that lies below the caudal end of the facial motor nucleus and extends up to 500 µm posterior to this level ($-11.6$ to $-12.8$ mm caudal to bregma; *Figure 3B–C*). This region contains the bulk of C1 and other blood pressure-regulating presympa-thetic neurons (*Dampney, 1994*; *Guyenet, 2006*; *Guyenet et al., 2013*). Smaller injections were performed to avoid the transduction of the retrotrapezoid nucleus (RTN) and the A1 region (*Fig-ure 3—figure supplement 1*; *Malheiros-Lima et al., 2018a*). *Figure 3B* shows the total number of TH$^+$ neurons expressing GFP in four rats used in the tracer experiments. On average, $56 \pm 3\%$ of the TH$^+$ neurons were immunoreactive (ir) for GFP (*Figure 3B*).

Transduced catecholaminergic neurons had putative catecholaminergic and glutamatergic vari-cosities within the pFRG (*Figures 2*, *3D–H*, *E'* and *H'*). We also noticed that every pontomedullary region that harbors noradrenergic neurons, including the locus coeruleus and the A1, A2 and A5 regions, also have GFP-labeled fibers (*Figure 3—figure supplement 1*). We observed (1) light pro-jections within the raphe nucleus, a region also noted for its role in respiratory control; (2) moderately dense projections to the nucleus of the solitary tract, which receives input from the carotid bodies and other cardiopulmonary afferents; and (3) extremely dense projections to the dorsal motor nucleus of the vagus, medullary raphe and RTN (*Figure 3—figure supplement 2*). All projections had a strong ipsilateral predominance.

## Parafacial respiratory group-projecting catecholaminergic neurons in the C1 region

Five cases in which CTb was injected within the pFRG were analyzed to visualize retrograde labeling in the C1 region (*Figure 4C and E*). A representative CTb injection centered on the pFRG of one rat is shown in *Figure 4B*. C1 neurons with projections to the pFRG were labeled with CTb 7–10 days before the rats were sacrificed. TH immunoreactivity was used to identify C1 neurons (*Figure 4C*

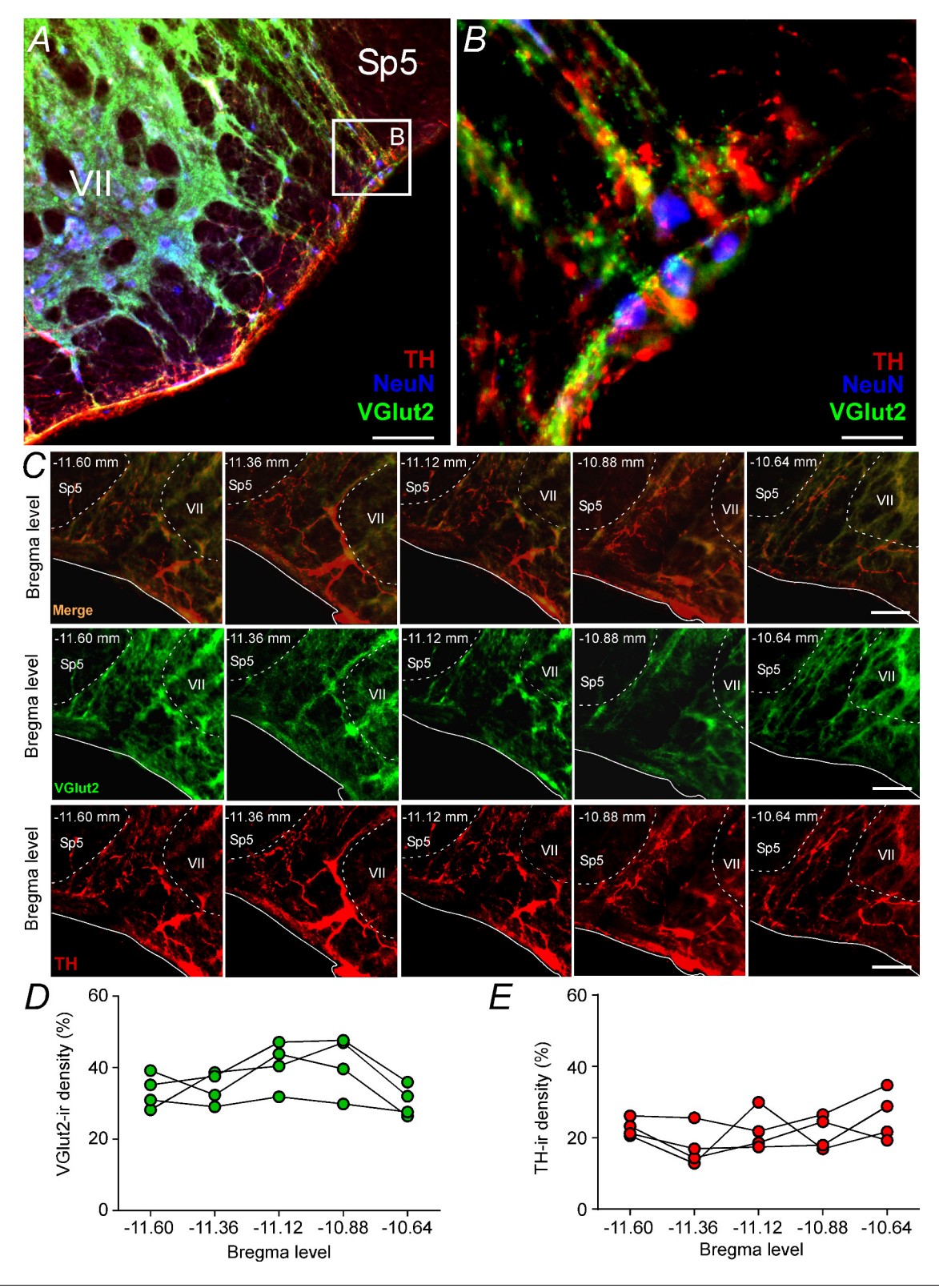

**Figure 2.** Glutamatergic and catecholaminergic innervations within the pFRG region. (A, B) Single case showing the close apposition of glutamatergic (VGlut2; green) and/or catecholaminergic (TH; red) terminals in the pFRG neurons (NeuN; blue). (C) Rostro-caudal distribution of glutamatergic (VGlut2; green) and catecholaminergic (TH; red) terminal in the pFRG region (Bregma level: −11.60 at 10.64 mm). Note the overlap between the VGlut2 and TH terminals. (D) VGlut2-immunoreactivity density in a rostro-caudal distribution (Bregma level from −11.6 to −10.64 mm). (E) TH-immunoreactivity density

*Figure 2 continued on next page*

Figure 2 continued

in a rostro-caudal distribution (Bregma level from −11.6 to −10.64 mm). Abbreviations: NeuN, nuclear neuronal marker; Sp5, spinal trigeminal tract; TH, tyrosine hydroxylase; VGlut2, vesicular glutamate transporter 2; VII, facial motor nucleus;. Scale bar in panel (A) = 1 mm, in panel (B) = 10 μm, and in panel (C) = 100 μm.

The online version of this article includes the following figure supplement(s) for figure 2:

**Figure supplement 1.** Total number of NeuN neurons located in the pFRG region.

and E). To characterize the C1 neurons that project to the pFRG, a series of 40-μm-thick coronal sections were analyzed, and cells were counted in only six levels of the C1 region, −11.6 to −12.80 mm relative to bregma, to include the lowest possible amount of catecholaminergic neurons in the A1 neurons. A considerably high number of retrograde labeled neurons (CTb$^+$) were found in the C1 region (*Figure 4C and E*), particularly in more rostral portions of the C1 region. For example, from a total of 129 ± 11 TH-ir cells in the C1 region, we found that 107 ± 10 neurons project to the pFRG, representing 83% of the TH cells in the C1 region (*Figure 4C and E*). As expected, CTb expression was also observed in cell bodies located in the commissural aspect of the nucleus of the solitary tract (cNTS); however, few double-labeled cells (49 ± 4 vs. total TH: 148 ± 9; 33% of TH cells in the A2 region) were found within the cNTS (*Figure 4D and F*).

## Parafacial respiratory neurons are activated by hypoxia

To verify whether neurons located in the pFRG region are activated by hypoxia, fos expression was evaluated in rats exposed to hypoxia (8% $O_2$ balanced with $N_2$; N = 6) or normoxia (21% $O_2$, 78% $N_2$, 1% $CO_2$; N = 6) for 3 hr. Coronal brainstem sections were double labeled for fos (used as a reporter of cell activation) and tyrosine hydroxylase (TH, a marker for catecholaminergic cell bodies and terminals). The number of fos and TH neurons were identified and counted in a one-in-six series of transverse sections (1 section every 240 μm). Counts were made on both sides of the brain and throughout the portion of the rostral aspect of the ventrolateral medulla (pFRG and C1 regions) (*Figure 5A–D and G–H*). We also counted the number of fos$^+$ neurons located in the cNTS (*Figure 5E–F and I*).

*Figure 5A* shows the pFRG region in a control rat (normoxia), whereas *Figure 5B* shows the pFRG region in a rat exposed to hypoxia for a period of 3 hr. Hypoxia caused a large increase in the number of fos$^+$ cells among presumably pFRG neurons (fos$^+$/TH$^-$) compared to that in normoxia (control group) (115 ± 14, vs. 7 ± 2 for normoxia; p<0.001) (*Figure 5A–B and G*). As expected, we also detected a large number of fos$^+$ cells among C1 neurons (fos$^+$/TH$^+$) (158 ± 11, vs. 14 ± 3 for normoxia; p<0.001) (*Figure 5C–D and H*). The number of fos$^+$ neurons in the cNTS (183 ± 8 vs. 10 ± 2 for normoxia; p<0.001) was higher in the hypoxia group than in the normoxia group (*Figure 5E–F and I*).

## Effect of bilateral blockade of ionotropic glutamatergic receptors in the parafacial respiratory group on active expiration elicited by chemoreflex activation

The experiment described below was designed to determine whether the activation of the pFRG eliciting active expiration during hypoxia or hypercapnia depends on glutamatergic synapses. Bilateral Kyn (100 mM, 50 nL) injections were performed in the pFRG (*Figure 6A–C*), and the changes in Dia$_{EMG}$, Abd$_{EMG}$ and AP that were induced by the effects of peripheral chemoreflex activation (bolus injection of KCN) were evaluated (*Figure 6D–I*).

As expected, the antagonism of glutamatergic receptors in the pFRG with Kyn injections did not alter the resting Dia$_{EMG}$ frequency (38 ± 5 bpm vs. 38 ± 6 bpm for baseline; p>0.05), Dia$_{EMG}$ amplitude (0.314 ± 0.114 mV vs. 0.264 ± 0.067 mV for baseline; p>0.05) or blood pressure (116 ± 7 mmHg vs. 120 ± 7 mmHg for baseline; p>0.05) (*Figure 6D, F–G and I*). Injections of Kyn into the pFRG did not evoke AE (*Figure 6D and H*).

Bilateral injections of Kyn into the pFRG did not change the increase in the Dia$_{EMG}$ frequency (58 ± 6 bpm vs. 71 ± 5 bpm for saline+KCN; p>0.05) induced by KCN (*Figure 6D and F*). However, blockade of ionotropic glutamatergic receptors in the pFRG reduced the Dia$_{EMG}$ amplitude (0.400 ± 0.163 mV vs. 1.714 ± 0.408 mV for saline+KCN; p<0.05) and completely eliminated the AE

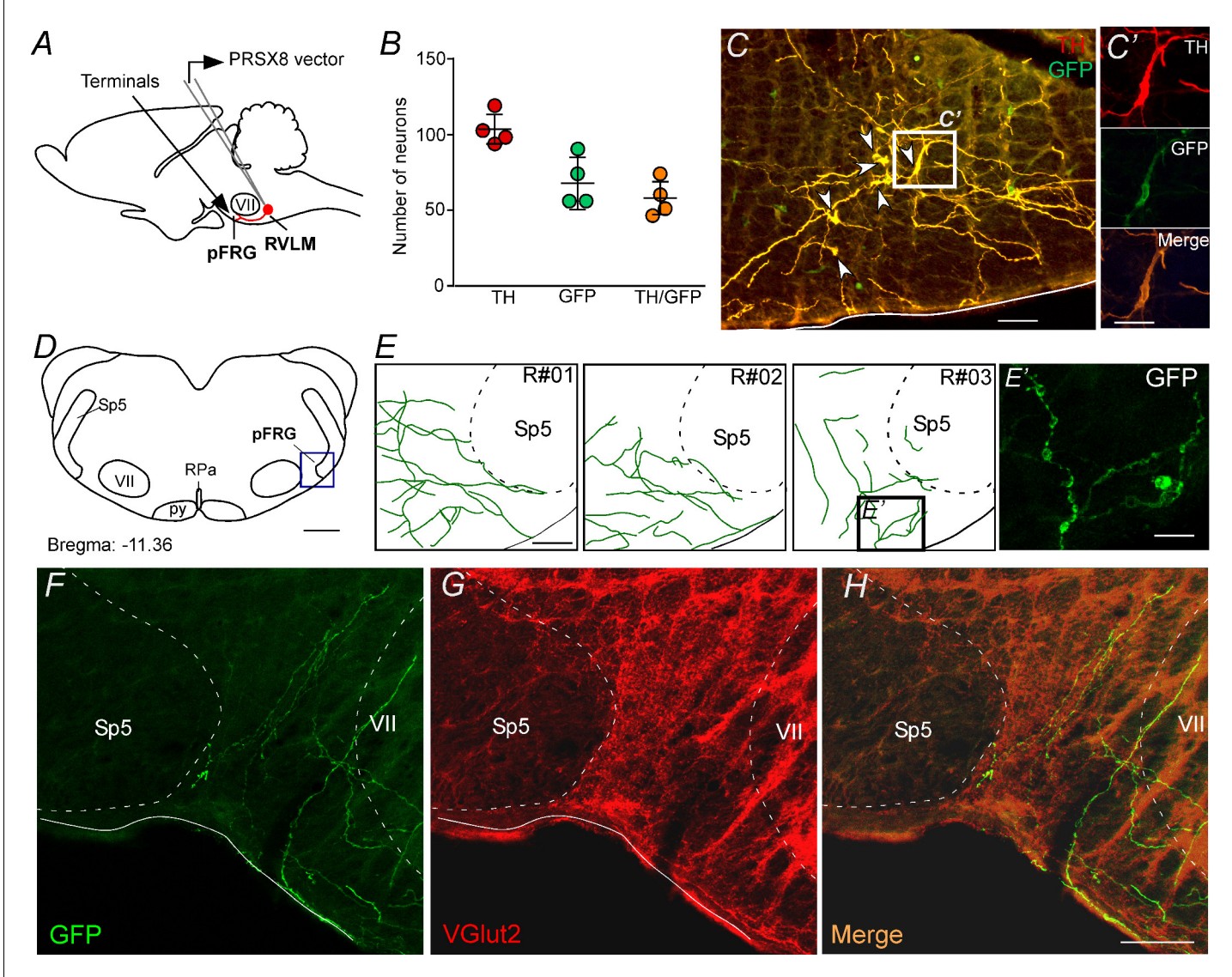

**Figure 3.** pFRG receives glutamatergic inputs from catecholaminergic C1 neurons. (**A**) Experimental design. (**B**) The number of TH and GFP neurons in the RVLM (N = 4). Cell count was obtained in six coronary brain sections (40 µm with 240 µm intervals between slices) from each rat. (**C, C'**) Single case showing that the C1 neurons are transfected by the lentivirus PRSX8-ChR2-eYFP. (**D, E**) Schematic drawing of a coronal brain section showing the distribution of GFP terminals in the pFRG region in three rats (bregma level: −11.36 mm in accordance with *Paxinos and Watson, 2007*). (**E'**) Photomicrography showing terminals into the pFRG region of the R#03. (**F–H**) Photomicrography showing fibers and terminals expressing lentivirus (GFP, green) and glutamate (VGlut2, red) in the pFRG region. Abbreviations: Fn, facial nerve; GFP, green fluorescent protein; pFRG, parafacial respiratory group; NTS, nucleus of the solitary tract; py, pyramid tract; RPa, raphe pallidus; RVLM, rostral ventrolateral medulla; Sp5, spinal trigeminal tract; TH, tyrosine hydroxylase; VGlut2, vesicular glutamate transporter 2; VII, facial motor nucleus. Scale bars in panels (C) and (E–H) = 50 µm, (C'), (E') and (H') = 20 µm, (D) = 1 mm.

The online version of this article includes the following figure supplement(s) for figure 3:

**Figure supplement 1.** C1 neurons project to the noradrenergic neurons.
**Figure supplement 2.** C1 neurons project to DVMN, medullary aphe and RTN regions.

generated by the $Abd_{EMG}$ induced by hypoxia (0.000 ± 0.000 mV vs. 0.386 ± 0.080 mV for saline +KCN; p<0.001) (*Figure 6D–E and G–H*). Kyn within the pFRG did not change the increase in AP elicited by KCN (142 ± 7 mmHg vs. 146 ± 9 mmHg for saline+KCN; p>0.05) (*Figure 6I*). At the same time, blockade of ionotropic glutamatergic receptors in the pFRG region did not change the increase in the $Dia_{EMG}$ frequency (54 ± 6 bpm vs. 68 ± 3 bpm for saline+KCN; p>0.05) or the $Dia_{EMG}$ amplitude (0.543 ± 0.285 mV vs. 1 ± 0.234 mV for saline+KCN; p>0.05) but completely eliminated

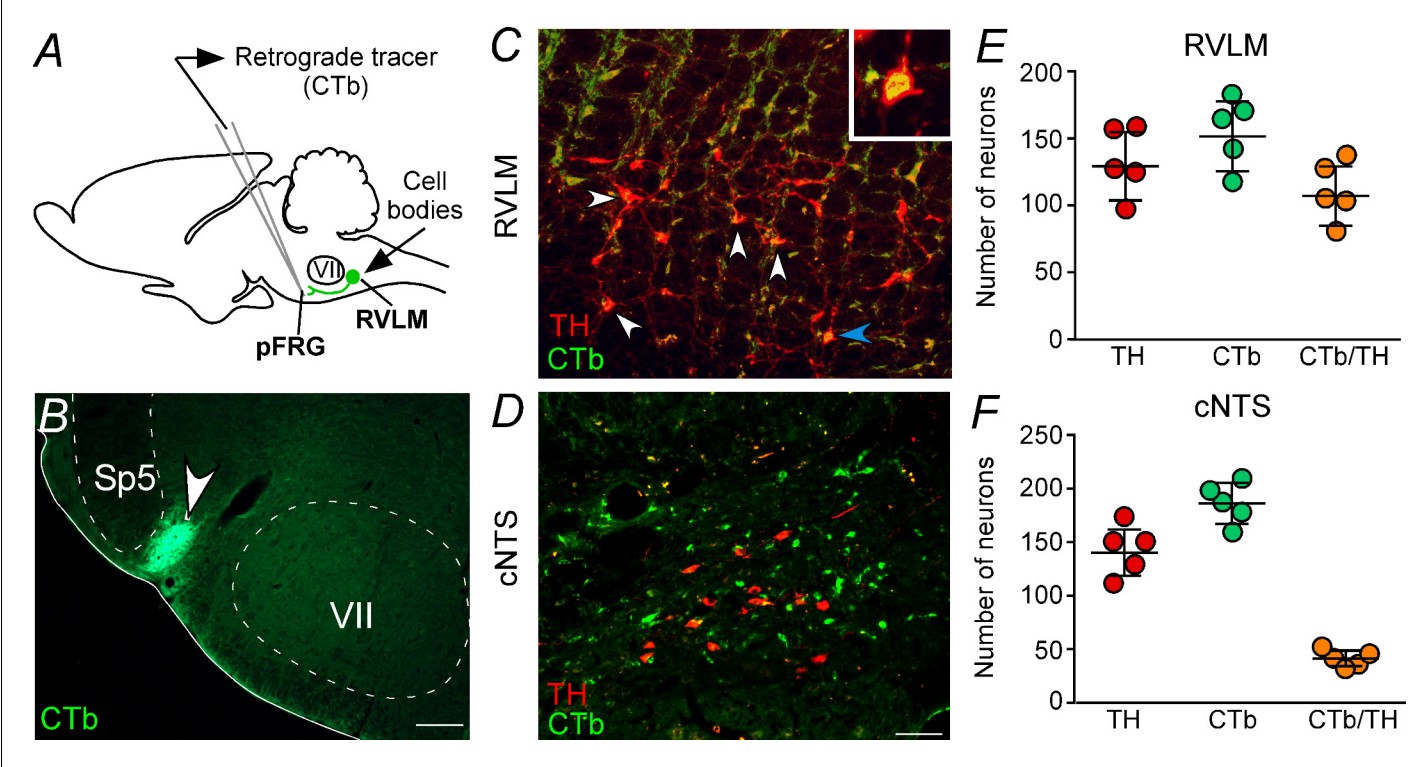

**Figure 4.** Catecholaminergic C1 neurons project to the pFRG region. (A) Experimental design. (B) Single case showing a positive injection of the retrograde tracer CTb (1%, 30–50 nL) into the pFRG region. (C, D) Photomicrography of catecholaminergic (TH, red) and CTb (green)-labelled cells located in the RLVM and cNTS of one rat with CTb-positive injection into pFRG. White arrows indicate some examples of double labeled cells, whereas the blue arrow indicates the high magnification double label (CTb$^+$/TH$^+$) in the RVLM region. (E, F) Number of CTb and/or TH-labeled cells in the RVLM and cNTS (N = 5). Cell count was obtained in coronary brain sections (six sections from the RVLM and three sections from the cNTS, each of 40 µm thickness with 240 µm of intervals between slices) from each rat. Abbreviations: cNTS, commissural nucleus of the solitary tract; CTb, Cholera Toxin b; pFRG, parafacial respiratory group; RVLM, rostral ventrolateral medulla; Sp5, spinal trigeminal tract; TH, tyrosine hydroxylase; VII, facial motor nucleus. Scale bars in panel (B) = 0.5 mm and in panel (D) = 50 µm (this scale bar applies to panels [C] and [D]).

the AE generated by Abd$_{EMG}$ induced by hypercapnia (0.000 ± 0.000 mV vs. 0.271 ± 0.042 mV for saline+KCN; p<0.001) (*Figure 6D and F–H*). Bilateral injections of Kyn into the pFRG did not change the increase in AP elicited by hypercapnia (*Figure 6I*).

## Changes in the active expiration elicited by chemoreflex activation after anti-DβH-SAP injections into the C1 region

The whole set of experiments performed in the present study demonstrated a possible role of C1 cells in the control of expiratory activity mediated by glutamatergic receptors at the level of the pFRG. The final experimental protocol aimed to evaluate the role of catecholaminergic neurons located in the RVLM in the expiratory activity elicited by hypoxia or by hypercapnia challenges. We used a toxin conjugated with DβH-SAP that was bilaterally injected into the rostral aspect of the ventrolateral medulla to produce a depletion of the TH-expressing neurons in the RVLM (*Figure 7A–C*). Only rats in which the intraparenchymal anti-DβH-SAP injections were confined to the RVLM were used. In the control rats, RVLM TH-ir profiles (presumably C1 cells) were found within the ventral aspect of all six of the brainstem levels examined (*Figure 7C*). Rats treated with anti-DβH-SAP had a depletion of the C1 neurons located in the RVLM (*Figure 7B–C*). In the six rats treated with anti-DβH-SAP, the counts of TH-ir between −11.60 and −12.80 mm relative to bregma revealed an average depletion of 71 ± 3% (range between 61–80%) (*Figure 7C*). At the intermediate and caudal medullary levels (12.80 to −15.20 mm caudal to bregma), the number of TH-ir neurons was unaffected by treatment with anti-DβH-SAP (158 ± 7 neurons vs. 169 ± 10 neurons for saline; p>0.05). The number of TH-ir neurons in the dorsal aspect of the medulla (A2 region) (307 ± 28

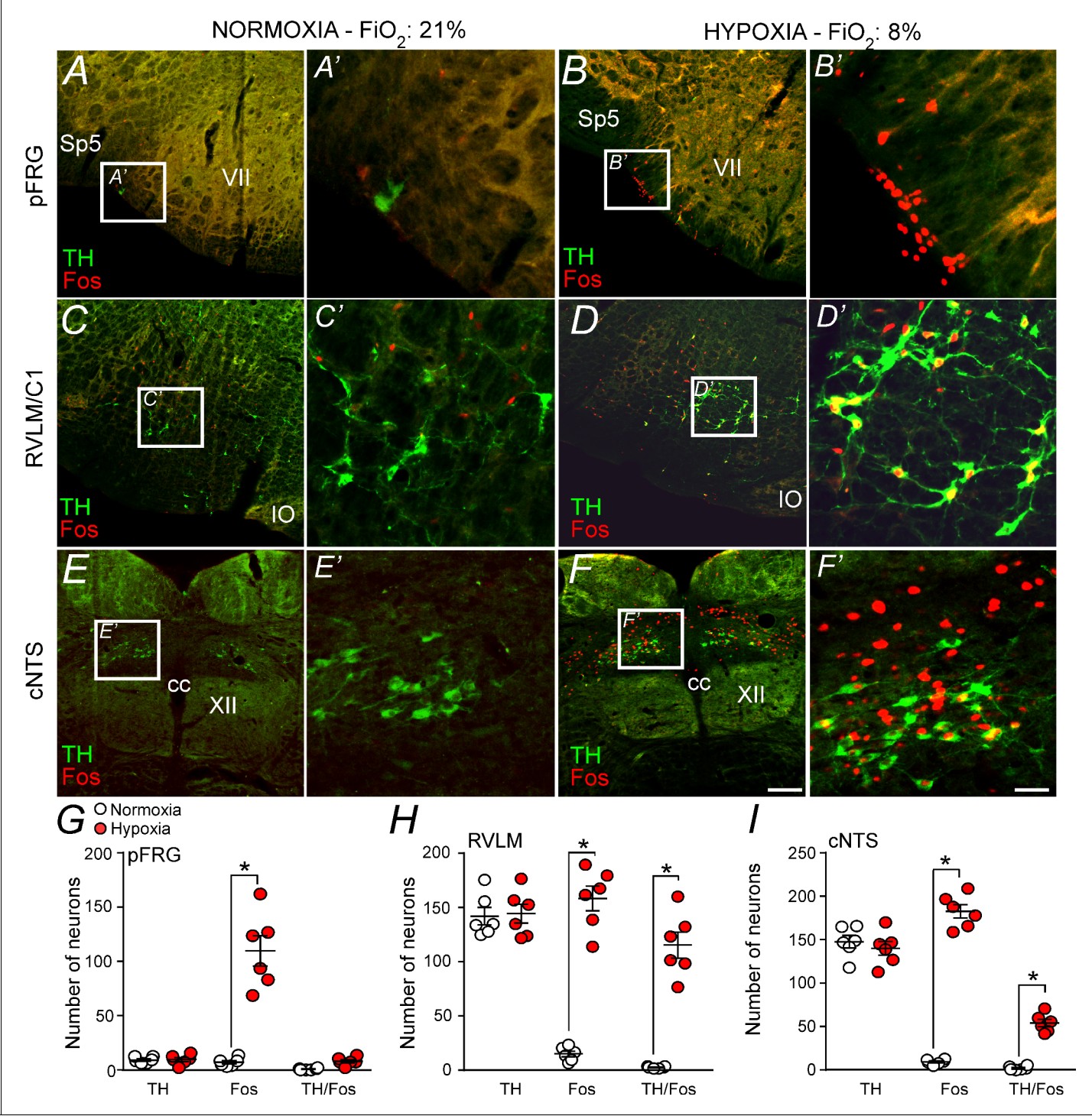

**Figure 5.** Hypoxia activates neurons in the pFRG, RVLM and cNTS. (A–F) Photomicrography showing representative cases of TH- and Fos-labelled cells located in(A, B) the pFRG, (C, D) the RLVM and (E, F) the cNTS after normoxia (21% $O_2$, balanced with $N_2$; N = 6) and after hypoxia (3 hr in 8% $O_2$, balanced with $N_2$; N = 6). (A'–F') Higher magnification of pFRG, RVLM and cNTS showing neurons labeled by Fos and/or TH. (G–I) Number of Fos and/or TH-labeled cells in the pFRG, RVLM and cNTS. Cell count was obtained in coronary brain sections (five sections from the pFRG, six sections from the RVLM and three sections from the cNTS of 40 μm in thickness with 240 μm intervals between slices) from each rat. Abbreviations: cc, central canal; cNTS, commissural nucleus of the solitary tract; IO, inferior olive; pFRG, parafacial respiratory group; RVLM, rostral ventrolateral medulla; Sp5, spinal trigeminal tract; TH, tyrosine hydroxylase; VII, facial motor nucleus; XII, hypoglossal nucleus. *Different from normoxia. Scale bar in panel (F) = 100 μm, this scale bar also applies topanels (A–F), whereas the scale bar in panel (F') = 20 μm applies to panels (A'-F').

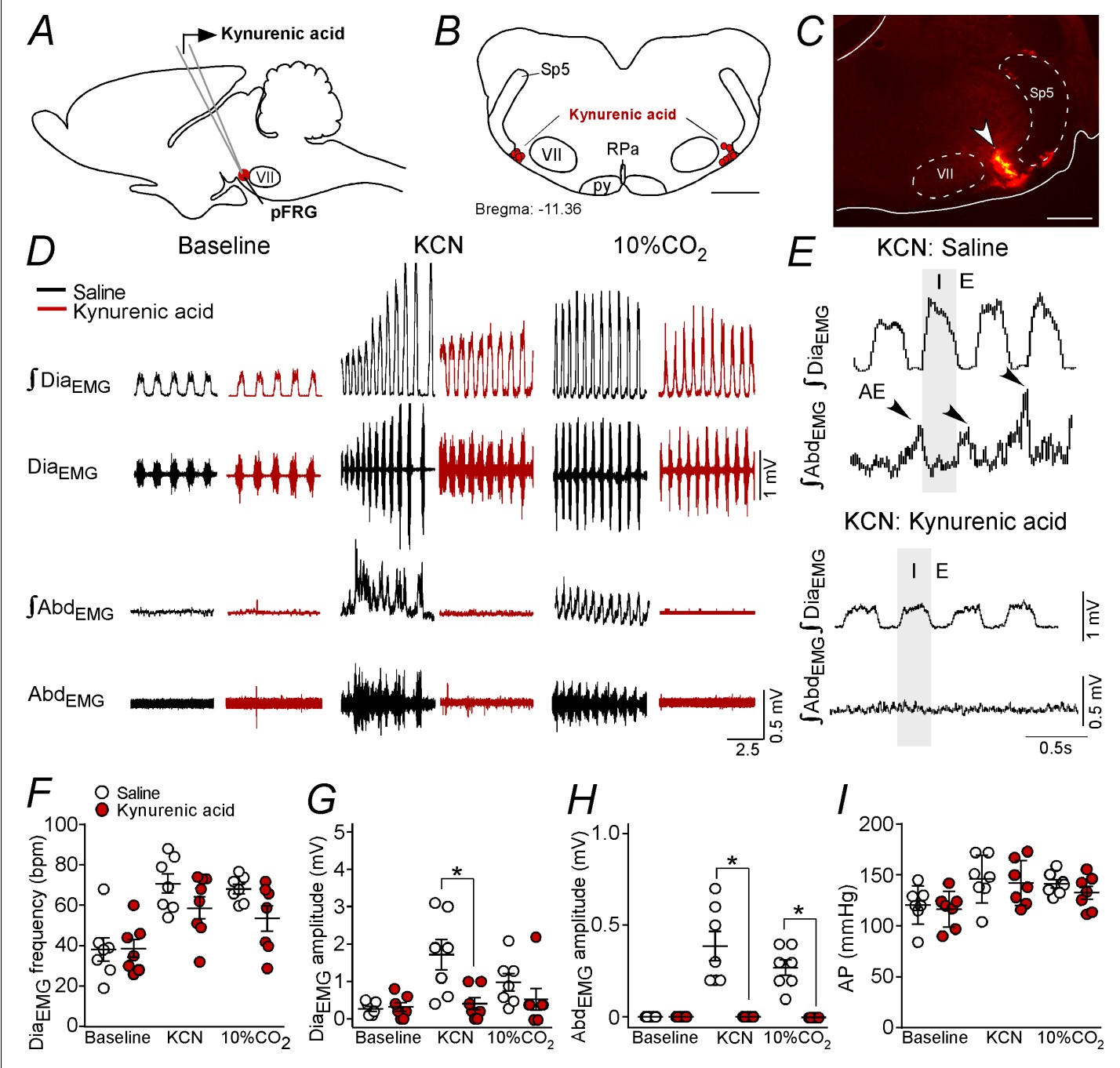

**Figure 6.** Kynurenic acid injection into the pFRG region blunted active expiration induced by hypoxia and hypercapnia. (**A**) Experimental design. (**B**) Bilateral sites for the injection of kynurenic acid (100 mM, 50 nL) into pFRG. (**C**) Photomicrography showing a typical site of kynurenic acid injection into pFRG. (**D**) Traces showing breathing parameters (diaphragm electromyography [$Dia_{EMG}$] and abdominal electromyography [$Abd_{EMG}$]) from a representative experiment after the injection of saline or kynurenic acid into the pFRG in a condition of cytotoxic hypoxia (KCN) or hypercapnia (10% $CO_2$). (**E**) Expanded traces showing the presence of active expiration (AE) (black arrow) after intravenous injection of KCN in a saline-treated rat and the absence of AE induction by KCN when ionotropic glutamatergic receptors are blockaded at the level of pFRG region. (**F–I**) Individual data for each parameter: (**F**) $Dia_{EMG}$ frequency (bpm), (**G**) $Dia_{EMG}$ amplitude (mV), (**H**) $Abd_{EMG}$ amplitude (mV) and (**I**) AP (mmHg) after bilateral injections of saline (white circles; N = 7) or kynurenic acid (red circles; N = 7) into pFRG under baseline, KCN or 10% $CO_2$ conditions. Lines show mean and SEM. *Different from saline; two-way ANOVA, p<0.05. Abbreviations: $Abd_{EMG}$, abdominal electromyogram; AP, arterial pressure; bpm, breaths per minute; $Dia_{EMG}$, diaphragm electromyogram; KCN, potassium cyanide; pFRG, parafacial respiratory group; py, pyriamidal tract; RPa, raphe pallidus; Sp5, spinal trigeminal tract; VII, facial motor nucleus. Scale bar in panel (**B**) = 1 mm and in panel (**C**) = 0.5 mm.

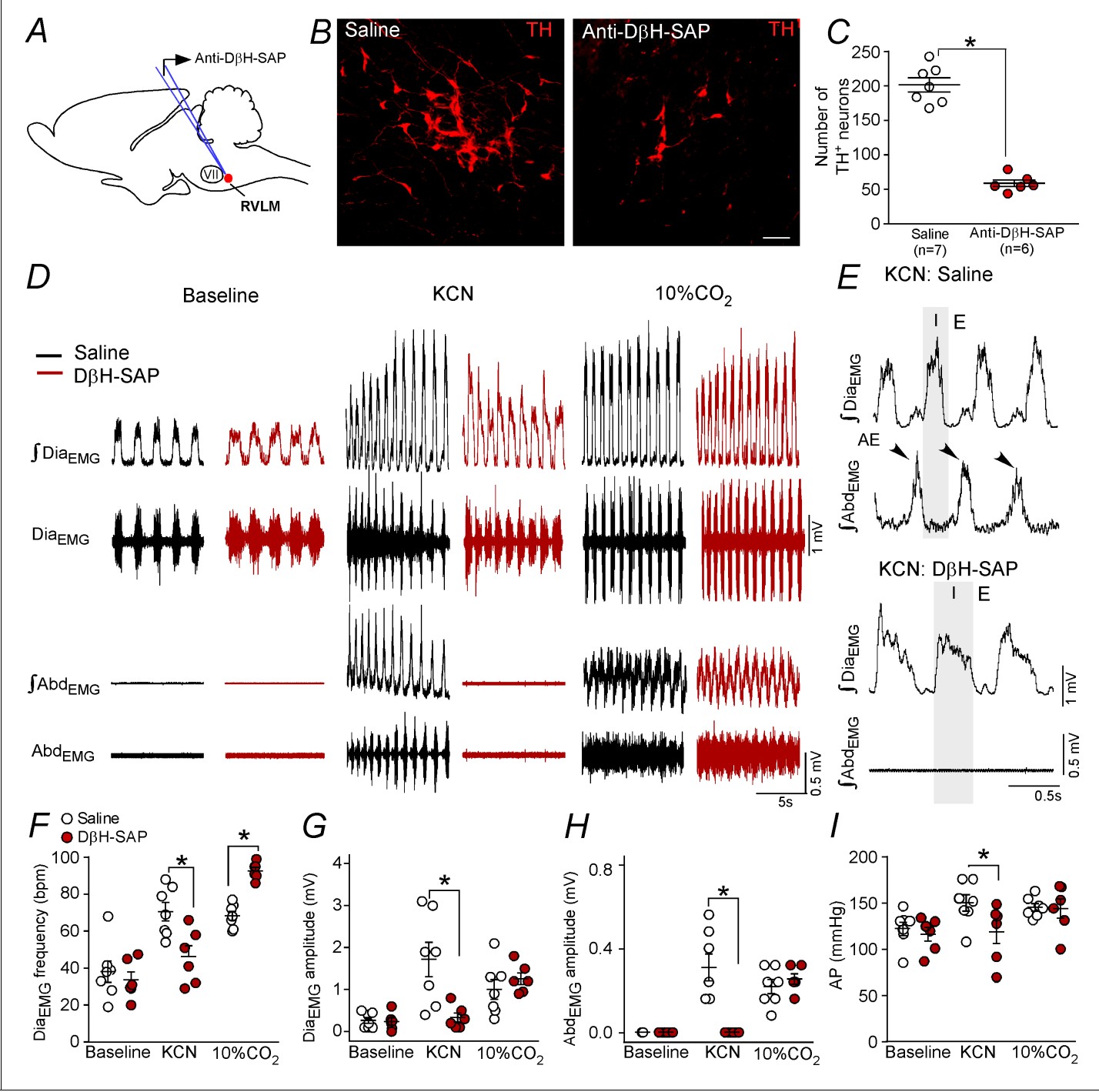

**Figure 7.** Ablation of the catecholaminergic C1 neurons blunted the active expiration induced by hypoxia, but not by hypercapnia. (A) Experimental design. (B) Photomicrography showing the C1 regions after bilateral injection of saline or anti-DβH-SAP (2.4 ng/100 nL) into the RVLM. (C) Number of TH neurons into the RVLM of the saline and anti-DβH-SAP-treated groups. (D) Traces showing breathing parameters (diaphragm electromyography [$Dia_{EMG}$] and abdominal electromyography [$Abd_{EMG}$]) from a representative experiment in a rat treated with saline or anti-DβH-SAP into the C1 region in a condition of cytotoxic hypoxia (KCN) or hypercapnia (10% $CO_2$). (E) Expanded traces showing the presence of active expiration (AE) (black arrow) after intravenous injection of KCN into a saline-treated rat and the absence of AE by KCN in an anti-DβH-SAP-treated rat. (F–I) Individual data for each parameter: (F) $Dia_{EMG}$ frequency (bpm), (G) $Dia_{EMG}$ amplitude (mV), (H) $Abd_{EMG}$ amplitude (mV) and (I) AP (mmHg) after bilateral injection of saline (white circles; N = 7) or anti-DβH-SAP (red circles; N = 6) into the RVLM under baseline KCN or 10% $CO_2$. Lines show mean and SEM. *Different from saline; two-way ANOVA, p<0.05. Abbreviations: $Abd_{EMG}$, abdominal electromyogram; Anti-DβH-SAP, immunotoxin anti-dopamine β-hydroxylase-

*Figure 7 continued on next page*

*Figure 7 continued*

saporin; AP, arterial pressure; bpm, breaths per minute; Dia$_{EMG}$, diaphragm electromyogram; KCN, potassium cyanide; RVLM, rostral ventrolateral medulla; VII, facial motor nucleus. Scale bar in panel (B) = 25 μm.

neurons vs. $331 \pm 30$ neurons for saline; p>0.05) or in the ventrolateral pons (A5 region) ($203 \pm 24$ neurons vs. $185 \pm 15$ neurons for saline; p>0.05) did not differ after the anti-DβH-SAP treatment.

Selective depletion of catecholaminergic neurons in the RVLM did not change breathing parameters at rest (*Figure 7D*; 7F–H). As expected, KCN or hypercapnia (10% $CO_2$ balanced with $O_2$) caused an increase in the Dia$_{EMG}$ frequency ($70 \pm 5$ bpm for KCN and $68 \pm 2$ bpm for $CO_2$ vs. $38 \pm 6$ bpm for baseline; p<0.05) and in the Dia$_{EMG}$ amplitude ($1.714 \pm 0.407$ mV for KCN and $1 \pm 0.234$ mV for $CO_2$ vs. $0.264 \pm 0.067$ mV for baseline; p<0.05), generated AE in the Abd$_{EMG}$ ($0.386 \pm 0.080$ mV for KCN and $0.271 \pm 0.042$ mV for $CO_2$ vs. $0.000 \pm 0.000$ mV for baseline; p<0.05) and increased AP ($146 \pm 9$ mmHg for KCN and $141 \pm 4$ mmHg for $CO_2$ vs. $120 \pm 7$ mmHg for baseline; p<0.05) (*Figure 7F–I*).

Depletion of catecholaminergic neurons in the RVLM blunted the increase in the Dia$_{EMG}$ frequency ($46 \pm 6$ bpm vs. $70 \pm 5$ bpm for saline; p<0.05), Dia$_{EMG}$ amplitude ($0.333 \pm 0.112$ mV vs. $1.714 \pm 0.408$ mV for saline; p<0.05), Abd$_{EMG}$ activity ($0.000 \pm 0.000$ mV vs. $0.386 \pm 0.08$ mV for saline; p<0.05) and pressor response ($146 \pm 9$ mmHg vs. $118 \pm 13$ mmHg for saline; p<0.05) elicited by KCN (*Figure 7D–I*). On the other hand, depletion of catecholaminergic neurons in the RVLM exacerbated the increase in the Dia$_{EMG}$ frequency ($93 \pm 2$ bpm vs. $68 \pm 2$ bpm for saline; p<0.05) (*Figure 7F*). Anti-DβH-SAP injected into the RVLM did not affect the Dia$_{EMG}$ amplitude ($1.258 \pm 0.139$ mmHg vs. $1 \pm 0.234$ mV mmHg for saline; p>0.05), Abd$_{EMG}$ activity ($0.316 \pm 0.031$ mV vs. $0.271 \pm 0.042$ mV for saline; p>0.05) or the pressor response ($141 \pm 10$ mmHg vs. $141 \pm 4$ mmHg for saline; p>0.05) elicited by hypercapnia (*Figure 7D–I*).

Injections of the toxin anti-DβH-SAP outside the C1 region often reached rostral aspects of the ventral lateral medulla, a region that harbors the chemically coded Phox2b$^+$/TH$^-$ neurons of the retrotrapezoid nucleus (N = 5) and facial motor nucleus (N = 2). The misplaced injections did not reach the caudal region of the ventrolateral medulla, that is the catecholaminergic A1 region. Bilateral injections of the toxin anti-DβH-SAP outside the C1 region produced no significant changes in the breathing parameters under normoxia, cytotoxic hypoxia or hypercapnia conditions (data not shown).

## Discussion

To summarize the results presented in this manuscript, we found a connection between catecholaminergic C1 neurons and the pFRG, which is essential for the generation of active expiration under hypoxic conditions. In addition, it seems that C1 neurons release glutamate at the level of the pFRG to trigger expiratory activity. Previous studies (*Marina et al., 2011*; *Abbott et al., 2013b*; *Burke et al., 2014*; *Malheiros-Lima et al., 2017*; *Malheiros-Lima et al., 2018b*; *Menuet et al., 2017*) and the present findings suggest that in the intact adult brain, C1 neurons recruit inspiratory and expiratory muscles in an orderly sequence that depends on the degree to which they are activated by hypoxia or by other mechanisms. Therefore, the present results are in agreement with Dr. Guyenet's theory suggesting that C1 cells are involved in an emergency body situation (*Guyenet, 2014*). In other words, these cells are recruited to help the organism to survive a major acute physical stresses.

### Catecholaminergic C1 neurons project to the pFRG

We found a significant number of fibers and varicosities within the pFRG region. Our transductions were restricted to the C1 region, indicating that GFP fibers and varicosities located in the pFRG belong to C1 neurons residing in the rostral aspect of the ventrolateral medulla. The strongest evidence that C1 neurons do in fact project to the pFRG is the presence of GFP and VGLUT2 double-labeling in this region. TH and CTb double-labeling in the C1 region, observed in rats that received retrograde tracer injection (CTb) into the pFRG, reinforces our hypothesis. Therefore, our data show that C1 neurons project directly to the pFRG and suggest that these neurons could modulate pFRG activity by releasing glutamate.

Catecholaminergic neurons from A2/C2 project to the parafacial region of the brainstem (*Viemari and Ramirez, 2006*; *Doi and Ramirez, 2010*). Therefore, the fibers and terminals labeled only with TH in the pFRG could derive from other catecholaminergic regions.

## Hypoxia triggering active expiration depends on C1 cells

Active expiration is defined as the recruitment of expiratory muscles for breathing that occurs during an increase of lung ventilation, for example during exercise, hypercapnia or hypoxia (*Iscoe, 1998*). Opinions vary regarding the role of the parafacial region (ventral x lateral subregions) in this process (*Janczewski and Feldman, 2006*; *Marina et al., 2010*; *Pagliardini et al., 2011*; *Abbott et al., 2014*; *Huckstepp et al., 2015*; *Huckstepp et al., 2018*; *de Britto and Moraes, 2017*; *Zoccal et al., 2018*; *Silva et al., 2019*). In addition, the anatomical origin, mechanism of generation and modulation of the breathing rhythm have been, and continue to be, intensely investigated (*Del Negro et al., 2018*). Our results add the information that the recruitment of active expiration under activation of peripheral chemoreceptors in anesthetized, vagotomized and artificially ventilated adult rats depends on the integrity of C1 cells.

As described before, active expiration may also be generated in hypoxic conditions (*Abdala et al., 2009*; *Huckstepp et al., 2015*; *Malheiros-Lima et al., 2017*). We propose the existence of a catecholaminergic mechanism for the generation of active expiration because selective ablation of catecholaminergic C1 neurons in the rostral ventrolateral medulla attenuated the late-expiratory flow and the late-E peak of $Abd_{EMG}$ activity observed during hypoxia in non-anesthetized rats (*Malheiros-Lima et al., 2017*) and in anesthetized, vagotomized and artificially ventilated rats (present study). This effect seems to be exclusive to the activation of peripheral chemoreceptors because ablation of the C1 cells did not affect the generation of active expiration elicited by the activation of central chemoreceptors (hypercapnia condition). These results suggest that the recruitment of abdominal muscles by chemosensory drives could be mediated through different neuromodulatory systems (*Leirão et al., 2018*; *O'Halloran, 2018*).

The control of expiratory activity by C1 neurons is consistent with their function in increasing lung ventilation according to the intensity of the hypercapnic or hypoxic stimulus (*Guyenet and Bayliss, 2015*; *Pisanski and Pagliardini, 2019*). The former and present data do not clarify whether inspiratory and expiratory activities are regulated by the same C1 neurons; however, active expiration triggered by hypoxia depends on the integrity of C1 neurons located in the rostral aspect of the ventrolateral medulla. The main question that needs to be addressed is how these neurons are involved in breathing regulation, especially under hypoxia. A direct connection between the C1 cells and elements of the ventral respiratory group has already been described (*Lipski et al., 1995*; *Agassandian et al., 2012*; *Abbott et al., 2013a*; *Burke et al., 2014*; *Stornetta et al., 2016*; *Menuet et al., 2017*; *Malheiros-Lima et al., 2018a*). C1 neurons are activated by hypoxia and then activate pFRG neurons through a direct glutamatergic mechanism to trigger active expiration. The hypoxic activation of C1 neurons is relayed by a direct input from the commissural nucleus of the solitary tract, which receives carotid body afferents (*Chitravanshi and Sapru, 1995*; *Aicher et al., 1996*; *Koshiya and Guyenet, 1996*; *Moreira et al., 2006*). The activation of C1 neurons by hypoxia can activate breathing through connections with the respiratory column, including the respiratory rhythm-generating neurons located in the pre-Bötzinger complex (*Pilowsky et al., 1990*; *Kang et al., 2017*; *Malheiros-Lima et al., 2018a*), and inspiratory premotor neurons in the rVRG (*Card et al., 2006*; *Guyenet et al., 2013*; *Burke et al., 2014*) and the pFRG region (present results). We should also consider other brainstem regions that are involved in breathing regulation, such as the parabrachial/Kolliker-Fuse complex (*Song and Poon, 2009*; *Damasceno et al., 2014*; *Silva et al., 2016a*), and the NTS (*Gozal et al., 1999*; *Song et al., 2011*).

As expected, we did not observe significant changes in AP after pharmacological manipulation of the pFRG region, which is consistent with the fact that the lateral aspect of the parafacial region has been proposed to be critical for expiratory rhythm generation only (*Janczewski and Feldman, 2006*; *Pagliardini et al., 2011*; *Huckstepp et al., 2015*; *Huckstepp et al., 2016*).

In our previous work, the depletion of C1 cells reduced the late-expiratory flow during hypoxia, suggesting that other mechanisms can contribute to generate active expiration in conscious adult rats (*Malheiros-Lima et al., 2017*). Although the C1 neurons express all the enzymes necessary to synthesize catecholamines, there is no direct evidence that they are able to use catecholamines as neurotransmitters (*Guyenet et al., 2013*). Until now, all previous studies showed strong evidence

that glutamate is required for the autonomic and respiratory effects mediated by the activation of C1 cells (*Abbott et al., 2014*; *Holloway et al., 2013*; *Guyenet et al., 2013*; *DePuy et al., 2013*; *Malheiros-Lima et al., 2018b*). In the present study, the blockade of ionotropic receptors in the pFRG and depletion of C1 cells eliminated the active expiration that was induced by activation of peripheral chemoreceptors. Our work, taken together with previous work from others, indicates that catecholaminergic neurons in the C1 region use glutamate primarily, if not exclusively, as a transmitter to influence downstream autonomic and/or respiratory neurons.

## Histology and experimental limitations

As shown previously (*Malheiros-Lima et al., 2018a*), most neurons that expressed GFP (94%) were catecholaminergic. The transduced catecholaminergic neurons were likely to be C1 (adrenergic) rather than A1 (noradrenergic) neurons based on: i) anatomical location, ii) expression of VGLUT2 immunoreactivity (*Stornetta et al., 2002*; *DePuy et al., 2013*; *Malheiros-Lima et al., 2018a*) and iii) catecholaminergic projections replicating the projections of the C1 cells as described by others (*Card et al., 2006*; *Menuet et al., 2017*; *Malheiros-Lima et al., 2018a*).

It is also important to note that GFP was expressed by Phox2b-containing, non-catecholaminergic (RTN-chemosensitive) neurons (*Stornetta et al., 2006*; *Abbott et al., 2009a*). However, very few transfected neurons were within the medullary surface under the facial motor nucleus, where almost every neuron is Phox2b-positive and TH-negative (*Takakura et al., 2014*).

Using optogenetic and pharmacological manipulation, we previously investigated whether glutamatergic/catecholaminergic neurotransmitters released from stimulated C1 cells contribute to cardiorespiratory parameters in urethane-anesthetized rats (*Malheiros-Lima et al., 2018b*). The selective activation of C1 cells in anesthetized, vagotomized, and artificially ventilated rats increased breathing frequency, but was not sufficient to induce an intense activation of breathing, and consequently to recruit active expiration (*Malheiros-Lima et al., 2018a*; unpublished results from our laboratory). We tried to perform a higher transduction of C1 cells; however, the PRSx8 promoter is also expressed in the RTN, making it a very difficult task to increase the number of C1 cells transduced by the lentivirus without a significant transfection of RTN chemoreceptor neurons.

There are strong and direct pieces of evidence that show that glutamate, but not catecholamines, determine the cardiovascular and respiratory responses evoked by activation of C1 cells (*Abbott et al., 2012*; *Abbott et al., 2014*; *DePuy et al., 2013*; *Guyenet et al., 2013*; *Holloway et al., 2013*; *Holloway et al., 2015*; *Malheiros-Lima et al., 2018a*). This evidence is also support by the absence of plasmalemmal monoamine transporter, which is necessary to replenish the catecholaminergic stores via reuptake in 90% of C1 cells (*Lorang et al., 1994*; *Comer et al., 1998*; *Guyenet et al., 2013*). In our study, the depletion of catecholaminergic neurons in the RVLM or the blockade of ionotropic glutamatergic receptors in the pFRG completely abolished active expiration induced by KCN. Although we do not have direct evidence to show that C1 cells release only glutamate into pFRG, on the basis of all previous data, we suggest that the short-term effects elicited by C1 neurons on active expiration may operate via ionotropic glutamatergic transmission.

We previously demonstrated that the depletion of C1 cells reduces the late-expiratory flow associated with active expiration during hypoxia in conscious rats (*Malheiros-Lima et al., 2017*), suggesting that other pathways act as a recurrent system to control active expiration. Contrary to what might be expected, in the present study, the blockade of glutamatergic receptors and the depletion of C1 cells eliminated the abdominal activity evoked by the activation of the peripheral chemoreceptors. This result suggests that glutamatergic signaling from C1 cells determines the control of active expiration, at least under anesthesia. KCN bolus injection activates peripheral chemoreceptors, but the physiological responses differ from those seen during environmental hypoxia exposure; thus, it is difficult to make a translational extrapolation from our data. Considering the experimental protocol differences, we suggest that under anesthesia, the activation of peripheral chemoreceptors using KCN induced a fast and intense increase in ventilation, which would explain the crucial importance of ionotropic glutamatergic signaling to trigger active expiration. In conscious rats, the hypoxic condition was sustained for 10 min, so that it is plausible to suggest that other mechanisms contribute to recruit and sustain the expiratory activity. One additional hypothesis is that the glutamate could be essential to produce a fast recruitment of the active expiration, with additional neurotransmitters contributing to maintaining expiratory activity. During prolonged exposure to hypoxia that was used to evaluate the c-fos staining, the metabolic, thermoregulatory and respiratory responses differed

from those seen under the previous protocols, which could result in different pattern of abdominal activity recruitment. In summary, we suggest that different excitatory transmitters could be released by different stimuli (physical exercise, sleep, hypoxia, hypercapnia, and so on), so that the importance of each excitatory or inhibitory transmitter is associated with the nature, intensity and duration of the stimulus, but this assumption still needs to be tested.

## Conclusion

In this study, we observed that C1 cells are activated by hypoxia and send projections to the expiratory oscillator located in the pFRG region to trigger active expiration. Once activated by hypoxia, C1 cells release glutamate to activate pFRG neurons and to generate active expiration, which is an emergent respiratory property that maintains physiological homeostasis when ventilatory demand is increased. Therefore, our study provides evidence that these two populations interact in a way that requires glutamatergic inputs from the C1 to the pFRG regions for the activation of expiratory neurons during hypoxia, which in turn is responsible for the generation of an active expiratory pattern. The revealed C1 and pFRG microcircuitry helps us to understand the neural organization of the respiratory pattern generator in conditions of peripheral chemoreceptor activation. Therefore, our findings have potential implications for understanding the developmental mechanisms that match respiratory supply and demand during hypoxia.

# Materials and methods

## Animals

Experiments were performed in 48 adult, male Wistar rats weighing 250–330 g. The animals were housed individually in cages in a room with controlled temperature ($24 \pm 2°C$) and humidity ($55 \pm 10\%$). Lights were on from 7:00 am to 7:00 pm. Standard Bio Base rat chow (Águas Frias, SC, Brazil) and tap water were available ad libitum. Animals were used in accordance with the guidelines approved by the Animal Experimentation Ethics Committee of the Institute of Biomedical Science at the University of São Paulo (protocol number: 07/2014).

## Viral tracer injection into the RVLM

We used a previously described lentiviral vector that expresses enhanced humanized channelrhodopsin-2 fused with eYFP under the control of the artificial Phox2b-specific promoter PRSx8 (pLenti-PRSx8-hChR2 (H134)-eYFP-WPRE, henceforth abbreviated PRSx8-ChR2-eYFP) as an anterograde tracer (N = 4) (*Hwang et al., 2001*; *Abbott et al., 2009b*; *Malheiros-Lima et al., 2018a*). Phox2b is a transcription factor expressed in subsets of brainstem neurons, including RTN and catecholaminergic C1 cells (*Pattyn et al., 1997*; *Stornetta et al., 2006*). The virus was produced by using the virus core located at the Institute of Cancer under the supervision of Dr. Bryan Strauss (University of São Paulo) and diluted to a final titer of $6.75 \times 10^{11}$ viral particles/ml with sterile PBS before injection into the brain. We verified that the above viral dilution produced specific transfection of catecholaminergic-expressing neurons. Under intraperitoneal (i.p.) injection of a mixture of ketamine (100 mg/kg) and xylazine (7 mg/kg) anesthesia, a glass pipette with an external tip diameter of 20 μm was inserted into the brain through a dorsal craniotomy. The lentivirus was delivered through the pipette by controlled pressure injection (60 PSI, 3–8 ms pulses). The stereotaxic coordinates for targeting C1 neurons were 2.8 mm caudal from lambda, 1.8 mm lateral from midline, and 8.4 mm ventral from the skull surface. The injection was targeted unilaterally to the RVLM in a volume of 150 nL (*Malheiros-Lima et al., 2018a*). Animals were maintained for no less than 4 weeks before they were used in anatomical experiments. The surgical procedures and virus injections produced no observable behavioral or respiratory effects, and these rats gained weight normally.

## Injection of a retrograde tracer into the parafacial respiratory group

In five animals, pFRG injections of the retrograde tracer cholera toxin b (CTb, 1% in deionized water; List Biological, Campbell, CA) were unilaterally performed. For pFRG injections, the dorsal surface of the brain was exposed via an occipital craniotomy. The stereotaxic coordinates for targeting the pFRG were 2.7 mm caudal from lambda, 2.6 mm lateral from midline, and 8.7 mm ventral from the skull surface. The CTb (30 nl) was delivered by pressure through glass pipettes. Retrograde tracers

were injected over 1 min, and the pipette remained in the tissue for at least 5 min to minimize movement of the tracer up the injection tract. The pipette was then removed, and the incision site was closed. The animals were allowed 7–10 days for surgical recovery and for transport of the retrograde tracer.

## Immunotoxin lesions

Surgical procedures were performed on rats anesthetized with an i.p. injection of a mixture of ketamine and xylazine (100 and 7 mg/kg of body weight, respectively). Postsurgical protection against infection included intramuscular injections of an antibiotic (benzylpenicillin, 160,000 U/kg). For selective chemical lesions of C1 cells, the rats were fixed to a stereotaxic frame, and the coordinates used to locate the RVLM (2.8 mm caudal from lambda, 1.8 mm lateral from midline, and 8.4 mm ventral from the skull surface) were based on the rat stereotaxic atlas (*Paxinos and Watson, 2007*). The tip of a pipette, connected to a Hamilton syringe, was inserted directly into the RVLM for bilateral injections of saporin conjugate-dopamine beta hydroxylase (Anti-DβH-SAP; Advanced Targeting Systems, San Diego, CA) (2.4 ng in 100 nL of saline per side; N = 6). On the basis of a previous publication from our laboratory and the present study, we did not notice any differences in neuroanatomical or physiological experiments in animals that received IgG-saporin or saline in the C1 region (*Taxini et al., 2011*; *Malheiros-Lima et al., 2017*). Thus, in the present study, the sham-operated rats were injected with saline (0.15 M; N = 7). After surgery, the animals were kept in recovery for 2 weeks before they were used in physiological experiments.

## Physiological preparation

A tracheostomy was performed under general anesthesia with 5% isoflurane in 100% oxygen. Artificial ventilation (1 mL/100 g, 60–80 breaths/min) was initiated with 3.0–3.5% isoflurane in pure oxygen and maintained throughout surgical procedures. The frequency of ventilation was adjusted as needed to maintain end-tidal $CO_2$ at the desired level. Arterial $PCO_2$ was estimated from measurements of end-tidal $CO_2$ and rectal temperature was maintained at 37 ± 0.5°C (*Guyenet et al., 2005*). This variable was monitored with a microcapnometer (Columbus Instruments). Positive end-expiratory pressure (1.5 cmH$_2$O) was maintained throughout to minimize atelectasis. The arterial pressure remained above 110 mmHg, and the diaphragm activity could be silenced by lowering end-tidal $PCO_2$ to 3.5–4.5%. In all animals, the femoral artery and vein were cannulated, and both vagus nerves were cut distal to the carotid bifurcation as previously described (*Malheiros-Lima et al., 2018a*). The diaphragm (Dia$_{EMG}$) and abdominal (Abd$_{EMG}$) muscles were accessed by a ventral approach, and muscle activities were measured before and after pharmacological manipulation within the pFRG region and chemoreflex challenges. Potassium cyanide (KCN: 40 μg/0.1 mL, intravenously [i.v.]) and 10% $CO_2$ (balanced with $O_2$) was used to activate chemoreceptors.

Upon completion of the surgical procedures, isoflurane was replaced by urethane (1.2 g/kg; i.v.) administered slowly. The adequacy of anesthesia was monitored during a 20 min stabilization period by testing for the absence of withdrawal responses, pressor responses, and changes in Dia$_{EMG}$ to a firm toe pinch. Approximately hourly supplements of one-third of the initial dose of urethane were needed to satisfy these criteria throughout the recording period.

## Physiological variables

As previously described, arterial pressure (AP), diaphragm muscle activity (Dia$_{EMG}$), abdominal muscle activity (Abd$_{EMG}$), and end-expiratory $CO_2$ (etCO2) were digitized with a micro1401 (Cambridge Electronic Design), stored on a computer, and processed off-line with version 6 of Spike two software (Cambridge Electronic Design, Cambridge, UK). Integrated diaphragm (∫DiaEMG) and abdominal (∫AbdEMG) muscle activities were obtained after rectification and smoothing (τ = 0.015 s) of the original signal, which was acquired with a 30–300 Hz bandpass filter. Dia$_{EMG}$ amplitude and frequency, Abd$_{EMG}$ amplitude and AP were evaluated before and after pharmacological manipulations and chemoreflex challenges.

## Drugs

All drugs were purchased from Sigma Aldrich (Sigma Chemicals Co.). Glutamate (10 mM, 50 nL; in sterile saline at pH 7.4; unilateral) and kynurenic acid, a nonselective ionotropic glutamatergic

antagonist (100 mM, 50 nL; first dissolved in 1 N NaOH and then diluted in phosphate-buffered saline at pH 7.4; bilateral), were pressure injected (Picospritzer III, Parker Hannifin) (50 nL in 3 s) through single-barrel glass pipettes (20 μm tip diameter) into the pFRG (*Malheiros-Lima et al., 2018a*). All drugs contained a 5% dilution of fluorescent latex microbeads (Lumafluor, New City, NY, USA) for later histological identification of the injection sites.

## Hypoxia protocol

Fos-like immunoreactivity evoked by hypoxia was studied in conscious, unrestrained adult rats. To acclimate the rats to the hypoxia environment prior to experimentation, they were kept in a plexi-glass chamber (5 L) that was flushed continuously with a mixture of 79% nitrogen ($N_2$) and 21% oxygen ($O_2$) at a rate of 1 L/min to allow them to become acclimated to the environmental stimuli associated with the chamber and to minimize unspecific fos expression. Rats were first acclimated for 45 min in the chamber and then subjected to acute hypoxia (8% $O_2$ balanced with $N_2$) or normoxic control (21% $O_2$) for 3 hr, as previously demonstrated (*King et al., 2013*; *Silva et al., 2016b*). At the end of the stimulus, the rats were immediately deeply anesthetized and transcardially perfused. All experiments were performed at room temperature (24–26°C).

## Histology

The rats were deeply anesthetized with pentobarbital (60 mg/kg, i.p.), then injected with heparin (500 units, intracardially) and finally perfused through the ascending aorta, first with 250 mL of phosphate-buffered saline (PBS, pH 7.4) and then with 500 mL of 4% phosphate-buffered paraformaldehyde (0.1 M, pH 7.4). The brains were extracted, cryoprotected by overnight immersion in a 20% sucrose solution in phosphate buffered saline at 4°C, sectioned in the coronal plane at 40 μm with a sliding microtome and stored in cryoprotectant solution (20% glycerol plus 30% ethylene glycol in 50 mM phosphate buffer, pH 7.4) at −20°C until histological processing. All histochemical procedures were completed using free-floating sections according to previously described protocols (*Malheiros-Lima et al., 2017*; *Malheiros-Lima et al., 2018b*).

For immunofluorescence experiments, we used the following primary antibodies: a) neuronal nuclei (NeuN) (mouse anti-NeuN antibody, 1:5000; Millipore, USA); b) tyrosine hydroxylase (TH) (mouse anti-TH, 1:1000; Chemicon, Temecula, CA, USA); c) ChR2-eGFP (chicken anti-GFP, 1:2000; Sigma, St. Louis, MO, USA); d) cholera toxin b (CTb) (goat anti-CTb, 1:1000; Chemicon, Temecula, CA, USA); e) vesicular glutamate transporter 2 (VGlut2, Slc17a6) (guinea-pig anti-VGlut2, 1:2000; Chemicon International, Temecula, CA, USA) and f) fos (rabbit anti-fos, 1:2000; Santa Cruz Biotechnology, CA, USA). All of the primary antibodies were diluted in phosphate-buffered saline containing 1% normal donkey serum (Jackson Immuno Research Laboratories) and 0.3% Triton X-100 and incubated for 24 hr. Sections were subsequently rinsed in PBS and incubated for 2 hr in a) murine blue donkey anti-mouse (1:500) for NeuN; b) Alexa488 or Cy3 goat anti-mouse (1:200) for TH; c) Alexa488 donkey anti-chicken (1:200) for eGFP; d) Alexa488 donkey anti-goat (1:200) for CTb; e) Alexa 488 or Cy3 goat anti-guinea pig (1:200; Molecular Probes, USA) for VGlut2 and f) Cy3 goat anti-rabbit (1:200; Molecular Probes) for fos immunostaining. All the secondary antibodies were from Jackson Laboratories (West Grove, PA, USA) unless otherwise stated. The sections were mounted on gelatin-coated slides in sequential rostrocaudal order, dried, and covered with DPX (Sigma Aldrich, Milwaukee, WI, USA). Coverslip were affixed with nail polish.

## Mapping

A series of six 40 μm transverse sections through the brainstem were examined for each experiment under bright field and epifluorescence using a Zeiss AxioImager A1 microscope (Carl Zeiss Microimaging, Thornwood, NY). Neurons immunoreactive for NeuN, TH, eGFP, VGlut2, CTb, and/or fos were plotted with the Stereo Investigator software (Micro Brightfield, Colchester, VT) utilizing a motor-driven microscope stage and the Zeiss MRC camera according to previously described methods (*Takakura et al., 2006*). Only cell profiles that included a nucleus were counted and/or mapped except in the cases where collaterals were mapped. The Stereo Investigator files were exported into the Canvas drawing software (Version 9, ACD Systems, Inc) for text labeling and final presentation. The neuroanatomical nomenclature is from *Paxinos and Watson (2007)*. Photographs were taken with a Zeiss MRC camera (resolution 1388 × 1040 pixels), and the resulting TIFF

files were imported into Canvas software. Output levels were adjusted to include all information-containing pixels. Balance and contrast were adjusted to reflect true rendering as much as possible. No other 'photoretouching' was performed. Figures were assembled and labeled within Canvas.

The total number of TH- (TH-ir), CTb- and/or fos-positive cells in the rostral ventrolateral medulla (RVLM: between 11.60 and 12.80 mm caudal to bregma level), commissural nucleus of the solitary tract (cNTS: between 14.28 and 14.76 mm caudal to bregma level), and parafacial respiratory group (pFRG: between 10.64 and 11.60 caudal to bregma level) were plotted as the mean ± SEM (RVLM: six sections/animal; cNTS: three sections/animal; pFRG: five sections/animal). The profile counts from the animals that received bilateral microinjections of anti-DβH-SAP reflected the sum of both sides of the brainstem and were compared with the counts from the control rats. The neuroanatomical nomenclature employed during experimentation and in this manuscript was defined by *Paxinos and Watson (2007)*. The profile counts from the animals that received PRSx8-ChR2-eYFP or CTb injections reflected only one side of the medulla and were compared with the counts from the control rats. Confocal images (Carl Zeiss, Jena, Germany) were used to evaluate the colocalization between axonal varicosities of eGFP and VGlut2 in the pFRG region. Terminal fields were mapped using an x63 oil-immersion objective by taking 0.3 µm z-stack images of both red and green fluorescence through tissue where discernible eGFP-labeled fibers were sharply in focus. These stacks were usually between 5 µm and 10 µm in depth. Terminals were marked as positive only when both eGFP and VGlut2 immunofluorescent profiles were in focus in at least two consecutive z-sections.

## Statistics

All statistics were performed using GraphPad Prism 6. The normality of the data was assessed using the Shapiro-Wilk test, and normally distributed data were expressed as the means ± SEMs. Statistical significance was assessed by Student's t-test or two-way ANOVA, with or without repeated measures, as appropriate. When applicable, the Bonferroni post hoc test was used. The significance level was $p < 0.05$.

## Acknowledgements

This work was supported by the São Paulo Research Foundation (FAPESP; grants: 2016/23281–3 to ACT; 2015/23376–1 to TSM) and the Conselho Nacional de Desenvolvimento Científico e Tecnológico (CNPq; grant: 408647/2018–3 to ACT). FAPESP fellowships were awarded to MRML (2014/07698-6 and 2017/08696–5 ) and to JNS (2014/23418-3), and CNPq fellowships were awarded to ACT (301219/2016–8) and to TSM (301904/2015–4). This study was also financed in part by the Coordenação de Aperfeiçoamento de Pessoal de Nível Superior - Brasil (CAPES) - Financial Code 001.

## Additional information

### Funding

| Funder | Grant reference number | Author |
| --- | --- | --- |
| Fundação de Amparo à Pesquisa do Estado de São Paulo | Graduate Student Fellowship | Milene R Malheiros-Lima |
| Fundação de Amparo à Pesquisa do Estado de São Paulo | 2016/23281-3 | Ana C Takakura |
| Fundação de Amparo à Pesquisa do Estado de São Paulo | 2015/23376-1 | Thiago S Moreira |
| Coordenação de Aperfeiçoamento de Pessoal de Nível Superior | Finance Code 001 | Thiago S Moreira |
| Conselho Nacional de Desenvolvimento Científico e Tecnológico | 301219/2016-8 | Ana C Takakura |

| Conselho Nacional de Desen-volvimento Científico e Tecno-lógico | 301904/2015-4 | Thiago S Moreira |
|---|---|---|

The funders had no role in study design, data collection, and interpretation, or the decision to submit the work for publication.

## Author contributions

Milene R Malheiros-Lima, Conceptualization, Data curation, Formal analysis, Investigation, Methodology; Josiane N Silva, Felipe C Souza, Data curation, Formal analysis, Investigation, Methodology; Ana C Takakura, Conceptualization, Resources, Supervision, Funding acquisition, Project administration; Thiago S Moreira, Conceptualization, Data curation, Formal analysis, Supervision, Funding acquisition, Validation, Investigation, Visualization, Methodology, Project administration

## Author ORCIDs

Thiago S Moreira (iD) https://orcid.org/0000-0002-9789-8296

## Ethics

Animal experimentation: This study was performed in strict accordance with the recommendations in the Guide for the Care and Use of Laboratory Animals of the National Institutes of Health. All of the animals were handled according to approved institutional animal care and use committee (IACUC) protocols (#07-2014) of the Institute of Biomedical Science of the University of São Paulo. All surgery was performed under anesthesia, and every effort was made to minimize suffering.

## Decision letter and Author response

Decision letter https://doi.org/10.7554/eLife.52572.sa1
Author response https://doi.org/10.7554/eLife.52572.sa2

## Additional files

### Supplementary files

• Transparent reporting form

### Data availability

All data generated or analyzed during this study are included in the manuscript.

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
