## [Decision Letter]

**Acceptance summary:**

This paper elegantly elucidates the role of noradrenergic C1 neurons in controlling active expiration during breathing. The insights gained in this study are novel and contribute to our understanding of cardiorespiratory integration in general. The authors have placed the work firmly in the context of integration of different behaviors and homeostatic functions by the CNS. Breathing is a dynamic behavior divided into three phases (inspiration, post-inspiration and active expiration. These three phases are the results of different muscles recruitment (in general: diaphragm, upper airways and abdominal muscles) and their rhythm are controlled by three distinct oscillators: pre-Bötzinger Complex driving inspiration; post-inspiratory complex driving post-inspiration and the lateral parafacial region (pFRG) driving active expiration. The authors verify anatomically that C1 neurons project to the pFRG region and demonstrate that blocking glutamatergic receptors at the pFRG level and depleting C1 neurons blunts the hypoxic activation of active expiration. A significant finding is that depletion of catecholaminergic neurons in the ventrolateral reticular formation where C1 neurons are located did not change breathing parameters at rest.

**Decision letter after peer review:**

Thank you for submitting your article "Adrenergic C1 neurons are part of the circuitry that recruits active expiration in response to hypoxia" for consideration by *eLife*. Your article has been reviewed by three peer reviewers, including Jan-Marino Ramirez as the Reviewing Editor and Reviewer #1, and the evaluation has been overseen by Ronald Calabrese as the Senior Editor. The following individual involved in review of your submission has agreed to reveal their identity: Robert Huckstepp (Reviewer #2).

The reviewers have discussed the reviews with one another and the Reviewing Editor has drafted this decision to help you prepare a revised submission.

Summary:

The present manuscript presents a sequence of experiments that provide evidence of an excitatory role of C1 cells in the brainstem as part of the circuitry involved in the generation of active expiration in hypoxia. Despite the fact that the authors approached related questions (in previous recent articles), the present manuscript presents useful new data. The neuronal processes involved in the generation of active expiration have been elucidated recently and the theme is in the central stage of neural control of breathing. Importantly, few studies have looked at active expiration during hypoxia exposure.

Essential revisions:

1) The authors verify anatomically that C1 neurons project to the RTN/pFRG region and demonstrate that blocking glutamatergic receptors at the RTN/pFRG level and depleting C1 neurons blunts the hypoxic activation of active expiration. The authors have not directly shown that C1 neurons release sufficient amounts of glutamate into the pFRG to elicit active expiration. Since the authors have shown that they are able to transfect 94% of C1 neurons with the PRSx8-ChR2-eYFP virus, the authors should be able to activate these neurons to show that they can induce active expiration and then inject kynurenic acid to block it. This would provide much stronger evidence of their hypothesis as it would show directly that glutamate release from the C1 neurons into the pFRG alters active expiration. The authors have already used both of these techniques in this manuscript and should be able to easily combine them. In this experiment after a rest period they could photoactivate the C1 neurons and use adrenergic blockers to test the catecholaminergic input as well. Please try to add this experiment within a two months period or if this is not possible, please provide a strong argument, why this has not been done and address this in the Discussion.

2) An important question is related to the fact that active expiration is completely suppressed by one isolated pathway/mechanism evaluated presently. This would suggest the absence of recurrent control systems that are well known in respiratory networks. For example, the blockade of ionotropic glutamate receptors in the pFRG completely eliminated active expiration elicited by KCN and hypercapnia. Is the glutamatergic transmission the only excitatory one existing onto the pFRG? There are some evidence in the literature supporting the view that cholinergic and serotonergic transmission are also a source of excitation at the pFRG level to elicited active expiration. Another example: Depletion of catecholaminergic neurons in the RVLM completely abolished active expiration induced by KCN. There is evidence showing A2 projections to pFRG. Moreover, the TH terminals labeled in the pFRG may derive from different catecolaminergic source (other than C1). Conversely, the present evidence strongly supports that, under the present experimental condition, a fraction of RVLM catecholaminergic neurons are needed to express active expiration in hypoxia exposure. Such a result may be circumstantial and specific to the animal model (anaesthetised rodent). Please chose to address the questions in the Discussion.

3) Along the same lines, since you did not always use hypoxia, but KCN bolus injections, please avoid "hypoxia" in the title and specify:.…"in response to peripheral chemoreceptors activation".

---

## [Author Response]

Essential revisions:1) The authors verify anatomically that C1 neurons project to the RTN/pFRG region and demonstrate that blocking glutamatergic receptors at the RTN/pFRG level and depleting C1 neurons blunts the hypoxic activation of active expiration. The authors have not directly shown that C1 neurons release sufficient amounts of glutamate into the pFRG to elicit active expiration. Since the authors have shown that they are able to transfect 94% of C1 neurons with the PRSx8-ChR2-eYFP virus, the authors should be able to activate these neurons to show that they can induce active expiration and then inject kynurenic acid to block it. This would provide much stronger evidence of their hypothesis as it would show directly that glutamate release from the C1 neurons into the pFRG alters active expiration. The authors have already used both of these techniques in this manuscript and should be able to easily combine them. In this experiment after a rest period they could photoactivate the C1 neurons and use adrenergic blockers to test the catecholaminergic input as well. Please try to add this experiment within a two months period or if this is not possible, please provide a strong argument, why this has not been done and address this in the Discussion.

Thank you for the excellent points and suggestions. We already used a similar protocol, combining the optogenetics and pharmacological manipulation to evaluate if the C1 cells activation increase breathing frequency by releasing glutamate and/or catecholamines into the pre-Bötzinger complex (Malheiros-Lima et al., 2018). We would like to emphasize two important points about the use of the same approach to evaluate if C1 cells contribute to increase active expiration using glutamate as neurotransmitter.

1) We previously performed the experiments suggested by the reviewers and we found some problems regarding this protocol in order to evaluate active expiration induced by C1 cells stimulation in our experimental condition. See Author response image 1 for a representative tracer from one rat, in which 74% of C1 cells from one site were transduced by the PRSx8-ChR2-eYFP virus. It shows the effect of C1 cells stimulation on cardiorespiratory parameters in urethane anesthetized rats. In conscious rats, it was previously demonstrated that the activation of C1 cells promotes an intense increase breathing frequency, tidal volume and minute ventilation (Abbott et al., 2014). Note that activation of C1 cells in anesthetized, vagotomized, and artificially ventilated rats is not enough to promote an intense activation of breathing, and consequently to recruit active expiration. We tried to perform higher transduction of C1 cells. However, the PRSx8 promoter is also expressed in the retrotrapezoid nuclei, making it impossible to increase the number of C1 cells transduced by the lentivirus without a significant transfection of RTN neurons, which are well known by their involvement with breathing control using glutamate as neurotransmitter. It is a huge challenge for us combining photostimulation of C1 cells and pharmacological manipulation of pFRG region in conscious rats. The distance between C1 and pFRG regions is too small (less than 200 μm) to allow the concomitant implantation of the fiber optic and the guide cannula (23G needle).

**Author response image 1. respfig1:** Cardiorespiratory effects elicited by optogenetic activation of C1 cells in urethane anaesthetized, vagotomized and artificially ventilated adult rats. The arterial pressure (AP); diaphragm electromyography activity (Dia_EMG_); and abdominal electromyography (Abd_EMG_) were measure before, during and after the 30 seconds of C1 photostimulation (474 nm blue laser).

2) There are strong and direct evidences that glutamate release, but not catecholamine, determines the cardiovascular and respiratory responses evoked by activation of C1 cells (Guyenet et al., 2014; Holloway et al., 2013; 2015; Abbot et al., 2012; 2014; DePuy et al., 2013; Malheiros-Lima et al., 2018). These evidences are also supported by the absence of plasmalemmal monoamine transporter, necessary to replenish the catecholaminergic stores via reuptake, in 90% of C1 cells (Guyenet et al., 2014; Commer et al., 1998; Lorange et al., 1994). We also would like to emphases that, specifically in our study, the depletion of catecholaminergic neurons in the RVLM or the blockade of ionotropic glutamatergic receptors in the pFRG completely abolished active expiration induced by KCN. Although we do not have a direct evidence showing that C1 cells release only glutamate into pFRG, we suggest, based in all previous evidences, that the short-term effects of C1 neurons on active expiration elicited by KCN may operate via ionotropic glutamatergic transmission.

Taking all this points into account, unfortunately, we are not able to add new experiments showing that C1 cells release glutamate into pFRG to produce active expiration. We estimate that would need at least 4-5 months to perform new experiments in conscious rats. Therefore, we add these caveats in the Discussion and hope that reviewers understand our decision.

2) An important question is related to the fact that active expiration is completely suppressed by one isolated pathway/mechanism evaluated presently. This would suggest the absence of recurrent control systems that are well known in respiratory networks. For example, the blockade of ionotropic glutamate receptors in the pFRG completely eliminated active expiration elicited by KCN and hypercapnia. Is the glutamatergic transmission the only excitatory one existing onto the pFRG? There are some evidence in the literature supporting the view that cholinergic and serotonergic transmission are also a source of excitation at the pFRG level to elicited active expiration. Another example: Depletion of catecholaminergic neurons in the RVLM completely abolished active expiration induced by KCN. There is evidence showing A2 projections to pFRG. Moreover, the TH terminals labeled in the pFRG may derive from different catecolaminergic source (other than C1). Conversely, the present evidence strongly supports that, under the present experimental condition, a fraction of RVLM catecholaminergic neurons are needed to express active expiration in hypoxia exposure. Such a result may be circumstantial and specific to the animal model (anaesthetised rodent). Please chose to address the questions in the Discussion.

We agree with the reviewers that this is a very important question. Studies from our group demonstrated that the blockade of ionotropic glutamatergic receptors into pFRG eliminates active expiration during hypercapnia in two experimental conditions: i) using anesthetized, vagotomized, and artificially ventilated rats (Silva et al., 2019) and ii) using in situ working heart-brainstem preparation (Zoccal et al., 2018). However, our previous work showed that the depletion of C1 cells reduces the late-expiratory flow associated with active expiration during hypoxia in conscious rats (Malheiros-Lima et al., 2018). Thus, we did not expect that the blockade of ionotropic glutamatergic receptors in the pFRG and the depletion of C1 cells would suppress the active expiration induced by KCN in anesthetized, vagotomized, and artificially ventilated rats. Therefore, we suggest that the magnitude of contribution of the glutamatergic signaling into the pFRG will precisely adjust the recruitment of abdominal muscles according to the lung ventilation demand, differing between hypoxic hypoxia and cytotoxic hypoxia. Another hypothesis is that the glutamate could be essential to initiate but not to sustain the active expiration for long periods. In addition, we suggest that other transmitters will be released by different stimulus (physical exercise, sleep, hypoxia, hypercapnia, etc.), being the importance of each excitatory or inhibitory transmitter associated with the nature and intensity of the stimulus. We added this information in the Discussion section.

3) Along the same lines, since you did not always use hypoxia, but KCN bolus injections, please avoid "hypoxia" in the title and specify:.…"in response to peripheral chemoreceptors activation".

The reviewers are correct, we modified the title, as suggested. Now it reads: “Adrenergic C1 neurons are part of the circuitry that recruits active expiration in response to peripheral chemoreceptors activation”.